# Spatial Autoregressive Model for Estimation of Visitors’ Dynamic Agglomeration Patterns Near Event Location

**DOI:** 10.3390/s21134577

**Published:** 2021-07-04

**Authors:** Takumi Ban, Tomotaka Usui, Toshiyuki Yamamoto

**Affiliations:** 1Department of Civil Engineering, Graduate School of Engineering, Nagoya University, Nagoya 464-8603, Japan; 2Faculty of Human Environments, University of Human Environments, Okazaki 444-3505, Japan; t-usui@uhe.ac.jp; 3Institute of Materials and Systems for Sustainability, Nagoya University, Nagoya 464-8603, Japan

**Keywords:** GPS, mobile phone data, spatiotemporal auto-regression model, adjacency matrix, travel behavior

## Abstract

The rapid development of ubiquitous mobile computing has enabled the collection of new types of massive traffic data to understand collective movement patterns in social spaces. Contributing to the understanding of crowd formation and dispersal in populated areas, we developed a model of visitors’ dynamic agglomeration patterns at a particular event using dynamic population data. This information, a type of big data, comprised aggregate Global Positioning System (GPS) location data automatically collected from mobile phones without users’ intervention over a grid with a spatial resolution of 250 m. Herein, spatial autoregressive models with two-step adjacency matrices are proposed to represent visitors’ movement between grids around the event site. We confirmed that the proposed models had a higher goodness-of-fit than those without spatial or temporal autocorrelations. The results also show a significant reduction in accuracy when applied to prediction with estimated values of the endogenous variables of prior time periods.

## 1. Introduction

### 1.1. Background

In recent years, several traffic accidents have occurred in the vicinity of certain public events. In such cases, participants’ movements do not follow normal traffic patterns, leading to an unexpected demand of transportation infrastructure. Thus, pedestrian or automobile traffic jams frequently occur. For example, during the Akashi pedestrian bridge accident of 2001, a stampede by a crowd of pedestrians caused a fatal accident [1]. It is likely that the pedestrian bridge became a bottleneck and unexpected congestion developed. Many accidents of pedestrian stampedes have also occurred outside Japan [2]. The Mina stampede of 2015 remains fresh in memory. Many deaths were caused from this accident, which occurred at an annual Haji pilgrimage.

To avoid such risks, visitors’ dynamic agglomeration patterns should be forecast in areas surrounding gathering event sites. There are numerous approaches for estimating pedestrians’ movement while at an event, and patterns of movement tend to differ before and after an event, as participants naturally gather and disperse.

On the contrary, the number of mobile phone users in Japan has been explosively increasing. According to the transition of contracts with mobile phone companies published by the Ministry of Public Management, Home, Affairs, Posts, and Telecommunications, the percentage of people with contracts reached 100% in 2012 [3]. Every mobile phone sold after 2007 has included GPS. Therefore, we can gather location data from almost all Japanese citizens by mobile phone rather than specialized GPS logger and IoT devices with high initial costs.

However, with respect to privacy, we can apply a coarse-grained modeling technique by aggregating mobile phone location data. The Konzatsu-Tokei^®^ dataset published by Zenrin DataCom, Co. Ltd. Examples of applications of this dataset include “Density Map” [4] and the “People Flow Project” [5]. In this study, we aim to understand unexpected traffic demand and construct a spatial autoregressive model for estimating the flow of people near event sites.

### 1.2. Literature Review and Problem Identification

Conventional travel demand analytics have typically utilized questionnaires or census data. This kind of data does not represent travel behavior in detail, although it might be possible additionally to obtain travel behavior data regarding the time period when the survey was conducted. In other words, this kind of data does not include real-time properties.

In response to these limitations, GPS data has often been used to understand real-time travel behavior. When utilizing GPS data, it possible to acquire detailed position and timestamp information of GPS devices, as well as information on the time the data was acquired. This data basically contains location information (latitude, longitude) and a timestamp. These contents are insufficient to analyze travel behavior, because some attributes such as mode of travel and purpose are not included. Therefore, GPS data usually used in combination with other data (ex. Census [6]).

On long-term time scales, the amount of GPS probe data available is generally too large to process effectively in resource-constrained research settings, as analysis of this kind of data requires substantial computational resources. Deep learning and machine learning approaches to solving such problems have recently attracted considerable attention [7,8,9,10].

Acquisition of GPS probe data generally requires excessively expensive equipment, often collected by moving vehicles. However, such data collection methodologies involve well-understood limits in terms of size and bias. Thus, mobile phone data has attracted research attention to avoid the limitations of other data collection methods. Mobile phones generally include several sensors, and data can be acquired from them such as call detail records(CDR) data, GPS data, and Bluetooth data.

Studies utilizing GPS data from mobile phone have examined origin-destination (OD) estimation [11,12,13], tourist travel behavior [14], recognition of daily living patterns [15,16,17,18], patten clustering of travel between home and work using GPS and CDR data [19], travel mode classification using mobile phone GPS and accelerometer signal data [20], among many others. GPS data is considered high-resolution information for accurately understanding object movement. Therefore, when analyzing long-term and large-scale travel behavior, one further pre-processing step (ex. map matching) is generally considered necessary.

CDR data is a transmission log between mobile phone and base stations. This data includes device location information and a timestamp recording each instance in which a mobile phone user performed a device action such as voice calls or data network access. Moreover, CDR data is less precise, but also more compact, occupying less storage space for a given time period compared to GPS data. Therefore, it is relatively more practical to process CDR data in large-scale or long-term analyses. Studies using CDR data have estimated OD information with Census data [21,22], as well as areas of congested traffic [23], and counts of railway passengers [24], and has been used to conduct activity-based analysis [25,26], and identify travel pattern by fusing CDR and Census data [27,28], among many other diverse applications.

However, CDR data can be used to determine identifying personal information like individuals’ home or work addresses if the head and tail of the acquired data is not removed. Thus, CDR data not only is of lower resolution and more compact size than GPS data, but also still involves significant privacy issues [29].

Aggregated population data which is preprocessed to address privacy issues has still more compact size than either CDR or GPS data from mobile phones. This data is already aggregated in a grid. Thus, this type of data is suitable for analyses of conducted over a long period of time or a wide geographical area. However, the details of individual trips (ex. origin, destination, purpose, etc.) are unclear in such data, and it should be handled carefully as a result. This type of data is often combined other data.

Many studies have been conducted concerning the characteristics of certain kinds of big data, for instance focusing on population estimation in disaster situations [30,31,32], or OD estimation [33]. Hayano et al. [30] analyzed aggregated population data during the Fukushima Nuclear Power Plant accident resulting from the 2011 Tōhoku earthquake and tsunami. This study clarified the public evacuation behavior. In order to understand such an evacuation, direct interview or questionnaires were required. The authors then visualized that behavior and estimated the population distribution in the area defined as an evacuation zone [33] updated home and workplace related OD matrices using aggregated population data, applying the entropy maximization principle to OD estimation. In Japan, an investigation known as the Person Trip survey is conducted once every ten years for the analysis of travel demand. The authors proposed a method for updating OD matrices using aggregated population information from mobile phone GPS data because OD matrices from the Person Trip survey were updated only with low frequency. However, few studies have applied the aggregated population data to travel demand estimation at smaller geographical spaces such as a crowd of participants in the vicinity of certain public events.

In contrast, research on travel behavior modeling focused on event participants at a higher level of granularity is being actively conducted. As discussed above, numerous studies have been conducted on analysis of travel behavior from a macro perspective. Some representative examples examined visitor behavior analysis by combining GPS and other data(survey data, questionnaire results, photo data) [34,35], identified activity patterns of theme park visitors using GPS [36]. Therefore, approaches to understanding recorded travel behavior using GPS data are beginning to attract attention. However, as yet, few studies have been conducted on estimation of travel behavior using mobile phone GPS data.

In terms of crowd behavior modeling (not only event participants), there are many approaches; neural network/deep learning approach [37], image processing to video monitoring data [38,39], indoor travel behavior analysis using Bluetooth tracking data [40], and extracting travel behavior using Wi-Fi data [41,42]. A high initial cost is required to acquire these data such as preparing cameras, receivers, and sensors. Also, the data are obtained only around installed location. On the other hand, GPS data from mobile phone is obtained more exhaustively, continuously, and at lower cost than those data. Therefore, GPS data from mobile phone is attractive for observing the crowd mobility.

Thus, many studies have been conducted on crowd mobility in events and population distribution estimation using mobile phone data such as CDR and GPS. To the best of our knowledge, few studies have been conducted on estimation of population movement in an area surrounding the event location using aggregated population data collected from mobile phones. In this study, we extract characteristics of aggregated population data near an event spot. Then, we develop and evaluate a spatial autoregressive model of population movement in the designated area and time period.

## 2. Methodology

### 2.1. Data

The aggregated population data used in this study is known as congestion statistical data and was collected by Zenrin DataCom, Co., Ltd. (Tokyo, Japan). This data was gathered using the “AUTO GPS” function of phones operating on the network of NTT DOCOMO Inc. (Tokyo, Japan), one of the largest mobile network companies in Japan. This function automatically transmits the location data at every 5 min from the mobile phones of those who allowed the company to collect the data. When mobile phones were out of range for GPS signal, device location data from cell towers were interpolated via GPS to estimate their position. The data collected in this process flow was then fit to a real population map of a region separated into a mesh grid. Thus, it became difficult to identify each individual user location due to the original and closed expansion coefficients.

In this study, we utilize this data from Nagoya, Japan. Table 1 shows an overview of the data used in this study. The research area is the whole of Nagoya city, which is separated into a grid approximately 2000 meshes of 250 m square each. This mesh size was set to ensure privacy and to reduce the effects of GPS-positioning errors.

The events treated in this study were professional baseball matches held on 7 and 13 July 2012, which had durations from 14:00 to 17:04 and from 18:00 to 21:57, respectively. The data was aggregated over a basic interval of 5 min. Therefore, data was collected from 14:00 to 17:00 for the first event, and from 18:00 to 21:55 for the second.

### 2.2. Descriptive Analysis

#### 2.2.1. Descriptive Analysis of the Whole Research Area

The aggregated population in each mesh is estimated by expansion factor to fits real population distribution. As mentioned above, this data was collected from mobile phones allowed by users. Therefore, this data may contain a self-selection bias, but it is not taken into consideration when expanded. To confirm the accuracy of this estimate, we compared it to the nighttime population of the same area, which was obtained from the national census held in 2010. However, zones of aggregation of nighttime population and aggregated population data naturally differ. Herein, we converted the zoning of the nighttime population and aggregated location data to the smaller zoning used by the Person Trip survey.

Figure 1 shows the result of a comparison between nighttime population and aggregated mobile device data. In Zone 602, the aggregated population was about 26 times larger than its nighttime value from the census. Zone 602 is a bustling street in Nagoya city; thus, there are many visitors even at night, so the aggregated population was much larger than the recorded nighttime population, because the aggregated population data was not limited only to the residents. On the other hand, in the suburban area (after Zone 1300), the aggregated population from the GPS data was about 1/10 smaller than the recorded nighttime value. It is possible that some users turned off their mobile phones when they were at home, and some error might have resulted from unit conversions, or from the possibility that children or elderly people without mobile phones were not counted in the aggregated population data.

In order to understand the movement patterns of people in the area during the event, we visualized the population in each mesh at 20:00, after a reasonable amount of time had passed following the start of the event, on 13 July (Figure 2). In the figure, darker-colored meshes indicate larger numbers of people. The people converged around the principal train station in this area (Nagoya Station) and the site of the event (Nagoya Dome).

#### 2.2.2. Descriptive Analysis around the Event Site

We limited our area of interest to the immediate vicinity of the event site and focused on detailed effects. The zones chosen for analysis included No. 1 to 4 (Figure 3). No. 1 included the event site and No. 4 included the nearest subway station (Nagoya Dome Mae Yada Station). Below, we present the results of the descriptive analysis conducted for each zone.

Figure 4 shows the estimated population in zone 1 at each hour. The blue and red lines in Figure 4 show the estimated population on 7 and 13 July, respectively. The blue and red dotted lines in this figure indicate the start and end times of each event. The transitions at these lines indicate that before opening, the participants were gathering gradually at the event site. During the event, the population at the event site remained largely in place, and then decreased sharply after the event ended.

Next, Figure 5 shows that the time series of population location data of people en route from the zones next to the event site to the nearest station (No. 2, No. 3, and No. 4) on 13 July. Around 10 p.m., a peak population moved to next mesh, with some time difference between grids.

Finally, we determined the average speed of population movement between zones. Table 2 shows a summary of peak movement times and estimated speeds population movement speeds. This estimated speed was calculated based the distance (250 m) between the centers of each zone. The farther the zone was from the event site, the faster the speed. The speed observed was close to conventional values representing average walking speeds (male: 4.75 km/h, female: 4.57 km/h) [45]. Therefore, the results indicate that the estimated population speed declined as a result of heavy congestion around the event site.

### 2.3. Modeling

According to basic analysis, the aggregated population data in the area surrounding the event spot shows that the population moved with some time differences. Hence, we constructed a spatial auto-regression model for estimation of visitors’ dynamic agglomeration pattern around the event site. In this study, we divided all data gathered on 7 and 13 July in half randomly. Then, we divided the dataset into periods before the event opening and after closing. Half of dataset was used to construct the model; the other half was then used to test it.

#### 2.3.1. Overview of the Model

Spatial auto-regressive models are well known and have been used in several recent studies [46,47,48]. We represented population movements within the target area over time as spatial and temporal variables in a spatial autoregressive model. Moreover, the directed adjacency matrix was assumed to have spatial dependence. In the employed spatial autoregressive model, the population in each zone is given by the following equation.
(1)$$\begin{array}{c} Y_{t}=\left\{\rho_{0}+\rho_{1}W_{1}+\rho_{2}W_{2}\right\}P+\beta_{0}+\beta X_{t}+\varepsilon_{t}\\ \varepsilon_{t}∼N\left(0,\sigma^{2}n\right) \end{array}$$
where $Y_{t}$ denotes the population of each zone at time *t*, P denotes the observed population of each zone at time *t* (excluding own mesh) or t−1 (including own mesh), $W_{1}$ and $W_{2}$ denote the 1st and 2nd adjacency matrices, respectively, $\rho_{0}$, $\rho_{1}$,and $\rho_{2}$ are the spatial lag parameters of zones A, B, and C, respectively (Figure 6), $\beta_{0}$ and β are the regression coefficients, $X_{t}$ is explanatory variables, and $\varepsilon_{t}$ is error term.

Usually, an adjacency matrix contains elements with values limited to “0” and “1”. If an element is “1”, the zone neighbors the target zone; otherwise, it does not. Each such matrix affects the others.

In this study, we utilized aggregated population data which was collected every 5 min. In this 5 min interval, participants of an event by definition could not move outside of a given mesh; the time resolution of the data was limited to this level of precision. Hence, we considered only 2-step adjacency matrices.

Variations in behavior were expected to be observed between movement patterns prior to the event and those collected afterwards. The difference was apparent based on the direction of aggregate movement flow. Prior to the opening of event venue, almost all flow was directed towards the event location. In contrast, after the event ended, the participants moved away from the event location. Thus, the conventional adjacency matrix is limited in its ability to describe these behaviors. Therefore, we propose a model separated by event time using directed adjacency matrices; one representing the time before the opening of the event venue and another after the closing of the event.

#### 2.3.2. Directed Adjacency Matrices

Generally, adjacency matrices are symmetric, which is useful when effects between neighborhoods are bidirectional. However, around the event spot, the flow of people takes on significant directionality. When before the event opened, the movement flow was directed towards the event spot. In contrast, after the event ended, the movement flow was directed away from the event location. Thus, these matrices should be asymmetric to reflect this changing directionality. Figure 7 shows the concept of a directed adjacency matrix.

#### 2.3.3. Area and Time Span for Model

Considering the effects of the event across the grid without focusing on the previous section, the analytical area was set to about 2.5 km square, not including the event site itself, because this model was designed to estimate the dynamics of the surrounding population. However, the proposed method considers 2-step adjacency matrices, the areas of which are padded by two blank rows and columns from the edge (Figure 8). In addition, the time span of analysis was set as the period from 2 h before the event to 2 h after the event.

#### 2.3.4. Explanatory Variables

Table 3 indicates the explanatory variables used in estimating values for the flow model. The “Shop dummy”, which is “0” or “1”, is referred to as “ProAtlas facility point data”. If there were some shops, restaurant, or the like in a certain zone, as determined by local digital maps from the ProAtlas series (published by Alps Mapping Co. Ltd. of Nagoya, Japan; currently unavailable), this dummy was set to “1”; otherwise, it was “0”. The distances to the event site and to the nearest station were defined as the straight-line distance between zones. We used the observed population on 21 July as the normal condition.

## 3. Results

The variables of each model were estimated under various assumptions. The results are shown in Table 4 and Table 5.

Comparison between Models 0 and 1 shows that the Akaike’s information criterion (AIC) was improved by spatial lags ($\rho_{1}$ = $\rho_{2}$). For Model 2, which considered the population in the target mesh at time t−1, AIC was further improved over its values in Models 0 and 1. Model 3, which described the spatial lag at time t−1, was found to have more interpretability than Model 1. Model 4 combined Models 2 and 3, allowing it to consider the population in the target mesh at time t−1 and the spatial lag of the neighboring meshes. Finally, Model 5 divided the spatial lags of Model 4 into two steps, which are described as $\rho_{1}$ and $\rho_{2}$. Model 5, which considered the population of the target zone at time t−1 and the 2-step spatial lags of the neighboring zones, had the best fit. In addition, Model 6 was constructed by replacing general adjacency matrices with directivity matrices. AIC, Model 6 had the lowest AIC among the 6 models considered.

The coefficients of population at t−1 and spatial lag of population at t−1 from between zone A and B (Figure 6) were significant. Each coefficient was positive. Prior to the opening of the event, people were gathering from around the event location. In the proposed method, the time window of data utilized is 5 min. Therefore, population at t−1 was highly correlated.

On the other hand, the coefficient of spatial lag of population at t−1 from zone C to B (Figure 6) was estimated to have a negative value (before event opening) and a positive value (after event closing); however both pattern were not found to be significant. It may be suspected that the population two zones away from the target zone gathered similarly not only to the target zone, but also to the adjacent zones. The coefficient of “Population in normal condition” was also significant and was estimated to be positive. Generally, population distribution cannot suddenly change within a single month. Therefore, the population was strongly influenced by population data representing normal conditions. Nevertheless, some coefficients were not significant. Overall, the models progressively improved in accuracy, although some parameters had to be reconsidered.

## 4. Discussion

RMSE (Root Mean Squared Error) was used to evaluate the model performance. RMSE is a commonly used as a measure of precision. Values close to 0 indicate that the model fits well. Figure 9 shows scatter plots of observed and estimated values of model 0 and Model 6 using the data prior to the event opening. Figure 10 shows scatter plots of observed and estimated values of Model 0 and Model 6 using the data from after the event closed. Table 6 shows a summary of correlations and RMSE.

The simple multiple regression model (Model 0) did not consider geometrical variables. Model 0 showed significant limitations in its capability to estimate population by combining normal and temporary behavior. However, the proposed spatial auto-regressive model with a directed adjacency matrix (Model 6) could consider geometrical relationships.

A comparison between Models 0 and 6 using data from before the event opening (Figure 9) and a comparison between these models using data from after the event closing (Figure 10) show that proposed model demonstrated improvement not only in terms of correlation but also of RMSE, similar to the trend observed in the above comparison in terms of AIC.

## 5. Conclusions

Aggregated population data was collected for each zone; thus individual movement was obscured and rendered more difficult to trace. In this study, a descriptive analysis of this data was performed at each mesh near the site of an event. The circumstances of people moving in the area surrounding the event location were revealed from this data by plotting time series. In addition, after the closing of the event, the peak timestamp of population movement near event location showed some delay. Based on these observed delays and geometrical relations between meshes, we confirmed the movement of event participants who left the event location.

To estimate the population around the event spot, we proposed a spatial autoregressive model using 2-step directed adjacency matrices. This model was applied to a 2.5 square kilometer area representing the vicinity of the event location. In this model, the effects on population movement at t−1 were divided into three parts (own mesh, next mesh, meshes two steps away away). Compared to a simple multiple regression model, the proposed model demonstrated a better fit to the data. Furthermore, we evaluated the differences between the estimated and actual population over time. These results confirm that the proposed model showed significant improvement over a simple multiple regression model, with a lower RMSE and higher correlation.

The model distinguishes congested meshes by a simple calculation using big data collected from mobile phones, though it may also be applied to more limited areas and time durations. The real-time predictions of the model also showed optimal correspondence with the true congestion situation.

An improved model could evaluate congestion areas with greater precision and consider the distribution of security personnel, as well as an increased number of subways and traffic regulations.

One of the limitations is the potential self-selection bias in the GPS data. The investigation on those who allowed/did not allow the company to collect the location data is the future research topic. The proposed model could estimate the population in the target mesh at *t* with the population around itself at t−1. The previous datapoint is set to 5 min before a given time *t* as the population at t−1. The coefficients of this model related to population 5 min prior and to the population in normal condition were significant. However, other coefficients were not significant. We selected this time interval to understand the travel behavior of event participants. Therefore, future work may focus on the impacts of variations in the time interval of the proposed model on its accuracy and applicability.

## Figures and Tables

**Figure 1 sensors-21-04577-f001:**
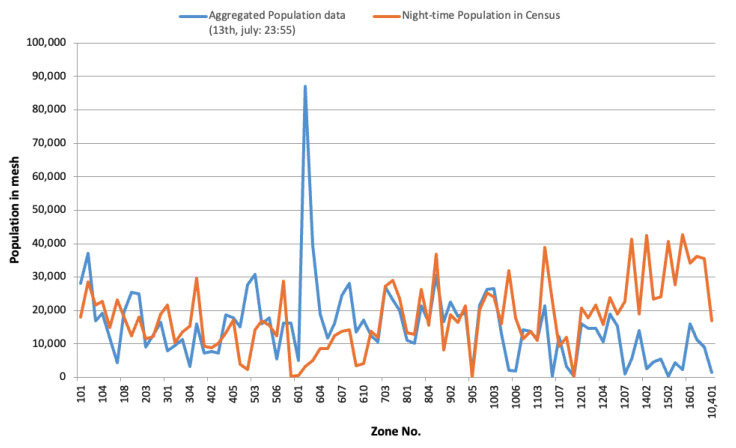
Comparison between aggregated population and nighttime population.

**Figure 2 sensors-21-04577-f002:**
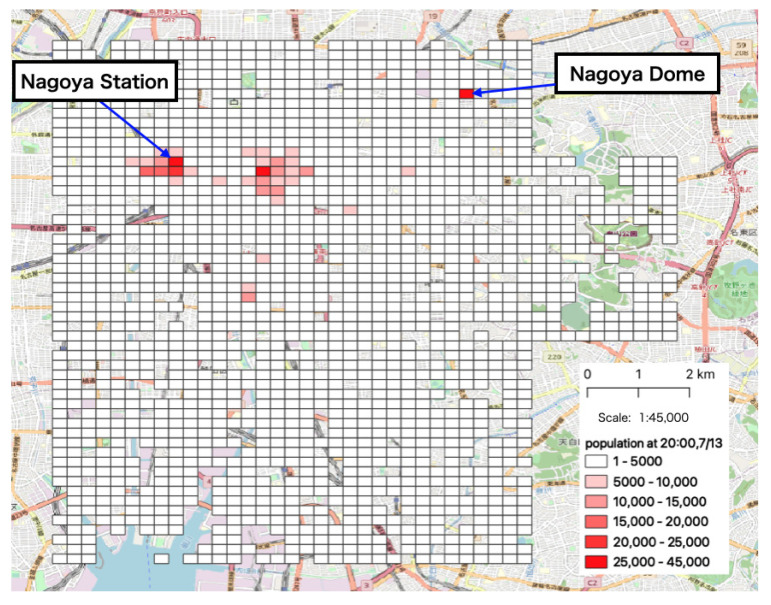
Visualization of the aggregated population data (13 July: 20:00).

**Figure 3 sensors-21-04577-f003:**
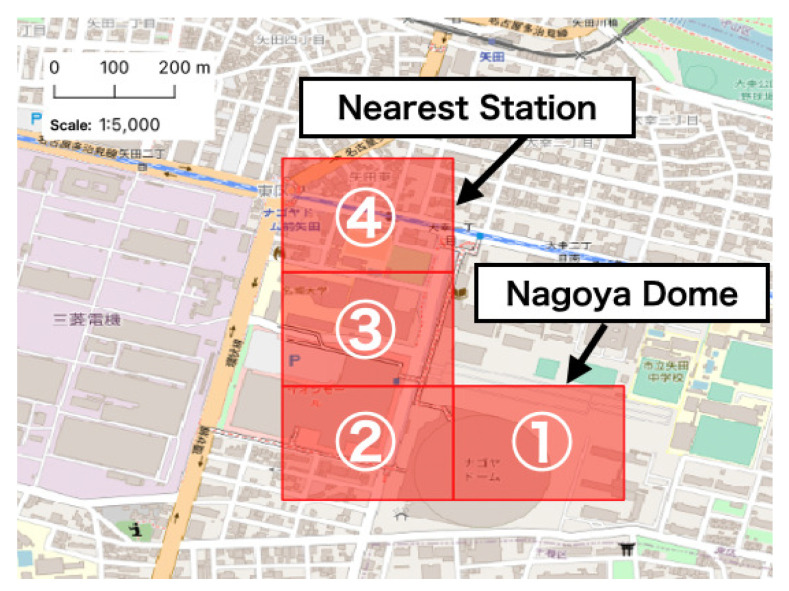
Four zones chosen for basic analysis near the event site (No. 1 includes the event site, Nagoya Dome, and No. 4 includes the nearest station to the event site).

**Figure 4 sensors-21-04577-f004:**
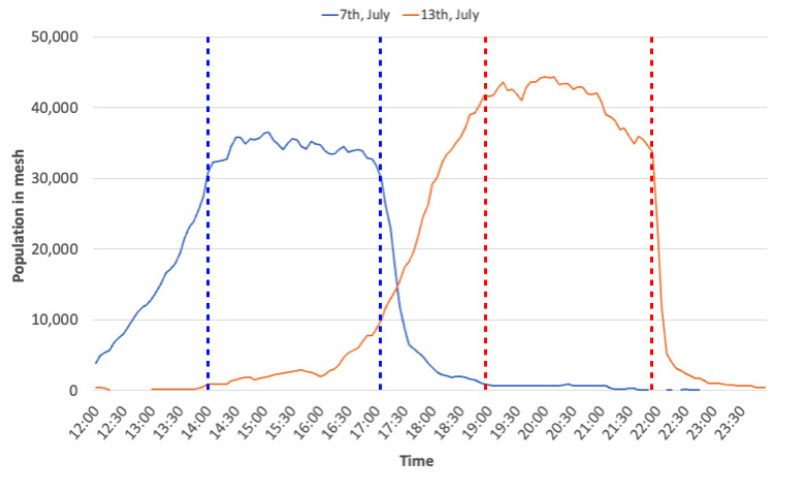
Time series of the observed population at the event site.

**Figure 5 sensors-21-04577-f005:**
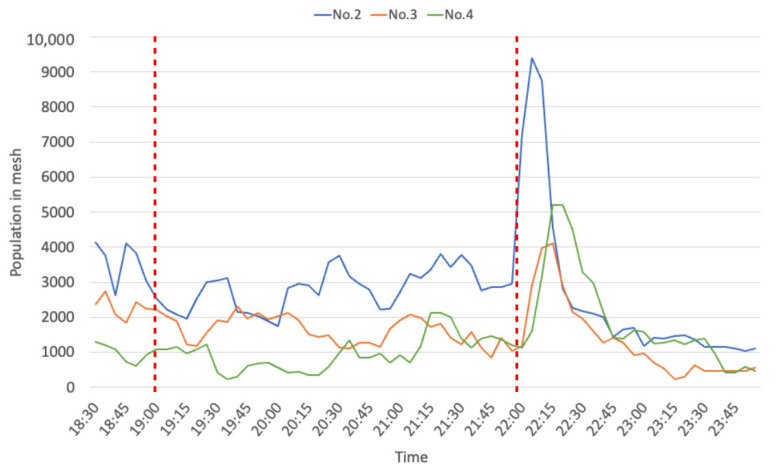
Time series of population movement en route to the nearest station (13 July).

**Figure 6 sensors-21-04577-f006:**
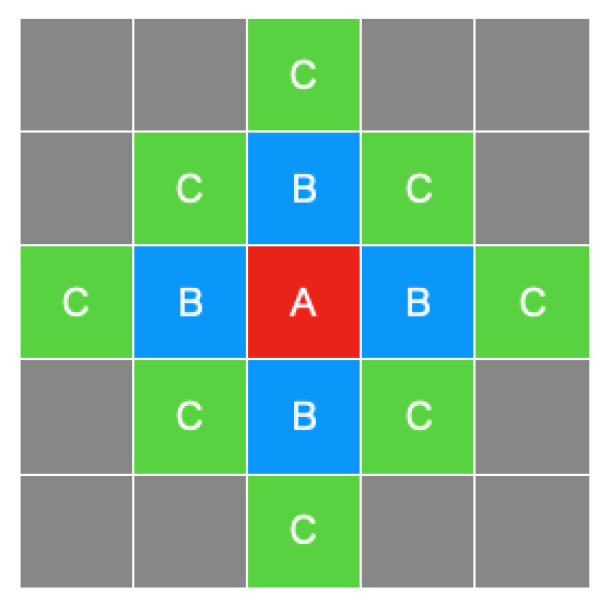
The positional relationship of neighboring zones.

**Figure 7 sensors-21-04577-f007:**
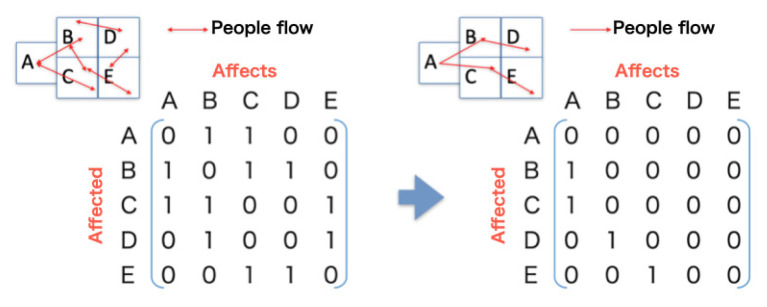
Concept of directed adjacency matrix.

**Figure 8 sensors-21-04577-f008:**
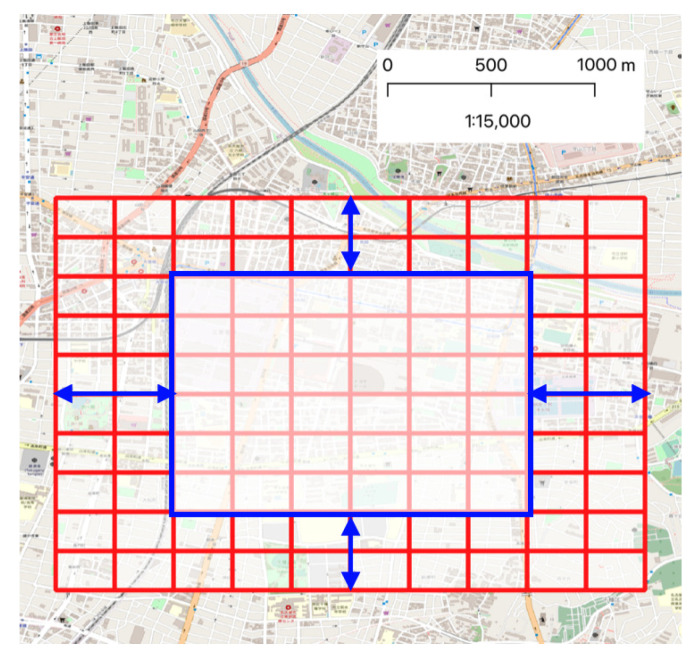
Analytical area.

**Figure 9 sensors-21-04577-f009:**
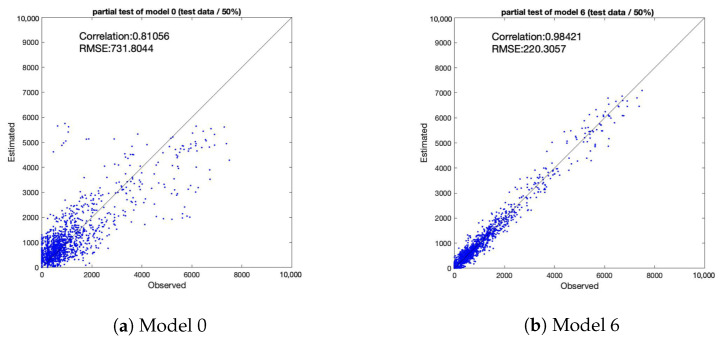
Correlation between observed and estimated population (before event opening).

**Figure 10 sensors-21-04577-f010:**
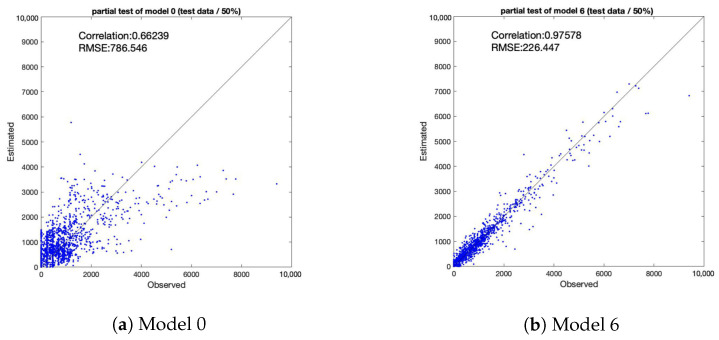
Correlation between observed and estimated population (after event closing).

**Table 1 sensors-21-04577-t001:** Overview of the data [43,44].

Data Item	Date/Time/Mesh Code/Population	
Acquisition Date	7 July 2012 (Sat.)	13 July 2012 (Fri.)
Time Slice	5 min	
Time Zone	12:00–23:55	
Aggregated Unit	250 m square mesh	
Type of Event	Baseball game	
Event Spot	Nagoya Dome	
Time for Event	14:00–17:04	18:00–21:57
# of Spectator	About 35,000	About 38,000

**Table 2 sensors-21-04577-t002:** Peak time and estimated speed (13 July).

Mesh No.	No. 1–No. 2	No. 2–No. 3	No. 3–No. 4
Peak Time	21:50→22:05	22:05 →22:10	22:10 →22:15
Time difference (min)	15	5	5
Estimated speed (km/h)	1.0	3.0	3.0

**Table 3 sensors-21-04577-t003:** Overview of explanatory variables.

Explanatory Variable	Unit	Detail
Shop dummy	-	In the target mesh, there is at least one shop; the value is “1”.
Time Difference from Event	min	Time difference from opening/closing time of event (before opening, negative; after closing, positive)
Distance to Event Spot	m	The straight distance to the event spot.
Distance to Station	m	The straight distance to the nearest station.
Event Scale	10,000 people	The number of spectators in an event.
Population in normal condition	person	The population in the target mesh on 21 July

**Table 4 sensors-21-04577-t004:** Results of models (using data before event opening).

			Model 0		Model 1		Model 2		Model 3		Model 4		Model 5		Model 6	
			W: Not Directed												W: Directed	
			$\rho_{0}=0$				$\rho_{0}\neq 0$		$\rho_{0}=0$		$\rho_{0}\neq 0$					
			$\rho_{1}=\rho_{2}=0$		$\rho_{1}=\rho_{2}$		$\rho_{1}=\rho_{2}=0$		$\rho_{1}=\rho_{2}$				$\rho_{1}\neq \rho_{2}$			
			$\rho_{1}^{\prime }=\rho_{2}^{\prime }=0$													
			**Coef.**		**Coef.**		**Coef.**		**Coef.**		**Coef.**		**Coef.**		**Coef.**	
Intercept			−1757	**	−1854	**	−252.6		−1851	**	−275.0		−272.2		−322.7	
P	$Y_{t-1}$	$\rho_{0}$	-	-	-	-	0.9359	***	-	-	0.9351	***	0.9313	***	0.9270	***
		$\rho_{1}$	-	-	-	-	-	-	0.01347	**	0.002912	*	0.007630	***	0.01315	***
		$\rho_{2}$	-	-	-	-	-	-	0.01347	**	0.002912	*	−0.0007092		−0.0009989	
	$Y_{t}$	$\rho_{1}$	-	-	0.001307	**	-	-	-	-	-	-	-	-	-	-
		$\rho_{2}$	-	-	0.001307	**	-	-	-	-	-	-	-	-	-	-
Shop dummy			63.80		73.76		15.89		74.16		18.17		17.76		19.74	
Time diff.			1.302	***	1.201	***	0.1108		1.193	***	0.08848		0.09618		0.08580	
Distance(event)			−496.1	***	−421.6	***	−13.53		−421.0	***	2.264		−8.219		−2.353	
Distance(station)			−71.35		44.41		−0.4857		49.12		25.50		28.89		44.48	
Event scale			640.0	**	608.7	***	74.42		606.1	***	67.58		69.08		77.94	
Normal pop.			0.9570	***	0.9493	***	0.06031	***	0.9491	***	0.05939	***	0.06615	***	0.06929	***
AIC			26,473		26,467		22,441		26,467		22,439		22,433		22,426	
Samples			1645													

***: 0.1% significant, **: 1% significant, *: 5% significant.

**Table 5 sensors-21-04577-t005:** Results of models (using data after event closing).

			Model 0		Model 1		Model 2		Model 3		Model 4		Model 5		Model 6	
			W: Not Directed												W: Directed	
			$\rho_{0}=0$				$\rho_{0}\neq 0$		$\rho_{0}=0$		$\rho_{0}\neq 0$					
			$\rho_{1}=\rho_{2}=0$		$\rho_{1}=\rho_{2}$		$\rho_{1}=\rho_{2}=0$		$\rho_{1}=\rho_{2}$				$\rho_{1}\neq \rho_{2}$			
			$\rho_{1}^{\prime }=\rho_{2}^{\prime }=0$													
			**Coef.**		**Coef.**		**Coef.**		**Coef.**		**Coef.**		**Coef.**		**Coef.**	
Intercept			4734	***	−1545	*	403.1		−1490	*	149.0		160.9		99.43	
P	$Y_{t-1}$	$\rho_{0}$	-	-	-	-	0.9260	***	-	-	0.9179	***	0.9145	***	0.9108	***
		$\rho_{1}$	-	-	-	-	-	-	0.0913	***	0.004490	*	0.008522	**	0.01083	***
		$\rho_{2}$	-	-	-	-	-	-	0.09125	***	0.004490	*	0.002066		0.003401	
	$Y_{t}$	$\rho_{1}$	-	-	0.08993	***	-	-	-	-	-	-	-	-	-	-
		$\rho_{2}$	-	-	0.08993	***	-	-	-	-	-	-	-	-	-	-
Shop dummy			15.81		54.72		9.569		55.27		11.56		8.795		11.32	
Time diff.			−1.743	***	0.1751		−0.1744	**	0.1574		−0.09451		−0.09956		−0.08190	
Distance(event)			−979.1	***	−364.5	***	−45.26		−350.4	***	−22.43		−23.13		−19.84	
Distance(station)			−144.9		249.9	**	5.622		264.4	**	24.46		24.48		37.82	
Event scale			−889.9	***	335.1		93.48		312.3	***	−41.24		−43.46		−29.94	
Normal pop.			1.149	***	1.117	***	0.07195	***	1.118	***	0.07979	***	0.08336	***	0.08576	***
AIC			30,502		30,181		26,209		30,165		26,205		26,205		26,201	
Samples			1873													

***: 0.1% significant, **: 1% significant, *: 5% significant.

**Table 6 sensors-21-04577-t006:** Summary of Correlation and RMSE.

	Before Opening		After Closing	
	Correlation	RMSE	Correlation	RMSE
Model 0	0.81	731.80	0.66	786.55
Model 6	0.98	220.31	0.98	226.45

## Data Availability

Restrictions apply to the availability of these data. Data was obtained from Zenrin DataCom, Co., Ltd. and are available from the authors with the permission of Zenrin DataCom, Co., Ltd.

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
