# Peer review of "Spatial Autoregressive Model for Estimation of Visitors’ Dynamic Agglomeration Patterns Near Event Location"

_sensors, 2021, doi:10.3390/s21134577_

Round 1

Reviewer 1 Report

The paper presents a spatial autoregressive model for predicting crowd dynamics near events. It clearly demonstrates the improved performances over a baseline multiple regression model. The data used are aggregate GPS positions of visitors. 

Major remarks:

The paper lacks a more rigorous comparison to related work. Namely:

1. The added value of the paper is demonstrated only in the context of predicting crowd dynamics using GPS positions. However, aggregate positions of visitors can be as well be provided with methods other than GPS, namely video monitoring, Bluetooth estimation, etc., and autoregressive models have been applied there, as well. It would be useful to provide a qualitative comparison to such approaches.

2. Also, there exist other methods that predict crowd dynamics based on GPS data, and also it would be useful to provide a qualitative comparison to those methods, if a quantitative comparison is not possible. In other words, the proposed method needs to be evaluated in the context of related state-of -art work, and not only compared to a baseline method.

For a recent survey of existing methods one can e.g. look at https://link.springer.com/article/10.1007/s10044-021-00959-z 

3. How does this approach addresses the fact that many users may turn on their GPS antenna only when actively using navigation maps, and otherwise turn it off. In other words, the visitors may be sometimes visible and sometimes invisible.

Minor remarks:

4. Not all variables in eq. (1) are explained

5. It is recommended to run the text through a grammar check, several prepositions are missing

Author Response

Dear Reviewer 1,

Thank you for providing these feedbacks you provided regarded our manuscript.

We summarized reply to your comments in attached file.

With these changes to our final manuscript, we hereby resubmit our manuscript for a secondary evaluation. Thank you once again for your consideration of our paper.

Sincerely,

Takumi BAN

Morikawa Yamamoto Miwa Lab., Department of Civil Engineering,
Graduate School of Engineering, Nagoya Univ.
Address: Furo-cho, Chikusa-ku, Nagoya 464-8603, Japan
Mail: [email protected]
TEL: +81-52-789-3565

Reviewer 2 Report

In the manuscript, the authors proposed a spatial autoregressive model using 2-step directed adjacency matrices to estimate the population around the event spot., the proposed model demonstrated a better fit to the data compared to a simple multiple regression model.

The topic is interesting, and the work is sufficient. To further improve the research work, the following revisions are suggested.

  • Quality of all figures are suggested to be improved.
  • More discussions on the advantages and shortcomings of the proposed method are suggested.
  • If possible, please re-organize the structure of the manuscript into the following typical structure, Introduction-Methods-Results-Discussion-Conclusion.

Author Response

Dear Reviewer 2,

Thank you for providing these feedbacks you provided regarded our manuscript.

We summarized reply to your comments in attached file.

With these changes to our final manuscript, we hereby resubmit our manuscript for a secondary evaluation. Thank you once again for your consideration of our paper.

Sincerely,

Takumi BAN

Morikawa Yamamoto Miwa Lab., Department of Civil Engineering,
Graduate School of Engineering, Nagoya Univ.
Address: Furo-cho, Chikusa-ku, Nagoya 464-8603, Japan
Mail: [email protected]
TEL: +81-52-789-3565

Round 2

Reviewer 2 Report

The manuscript has been well revised.